# Coarse Pruning of Convolutional Neural Networks with Random Masks

**Sajid Anwar, Wonyong Sung**
Department of Electrical Engineering and Computer Science
Seoul National University
Gwanak-Gu, 08826, Republic of Korea
`sajid@dsp.snu.ac.kr, wysung@snu.ac.kr`

## Abstract

The learning capability of a neural network improves with increasing depth at higher computational costs. Wider layers with dense kernel connectivity patterns further increase this cost and may hinder real-time inference. We propose feature map and kernel pruning for reducing the computational complexity of a deep convolutional neural network. Due to coarse nature, these pruning granularities can be exploited by GPUs and VLSI based implementations. Further, we propose a simple strategy to choose the least adversarial pruning masks. The proposed approach is generic and can select good pruning masks for feature map, kernel and intra-kernel pruning. The pruning masks are generated randomly, and the best performing one is selected using the validation set. The sufficient number of random pruning masks to try depends on the pruning ratio, and is less than 100 when 40% complexity reduction is needed. Once the least adversarial pruning mask is selected, we prune and retrain the network in one-shot. The proposed approach therefore consumes less time compared to iterative pruning. We have extensively evaluated the proposed approach with the CIFAR-100, CIFAR-10, SVHN, and MNIST datasets. Experiments show that 60-70% sparsity can be induced in the convolution layers with less than 1% increase in the misclassification rate of the baseline network.

## 1 Introduction

Deep and wider neural networks have the capacity to learn a complex unknown function from the training data. The network reported in Dean et al. (2012) has 1.7 billion parameters and is trained on tens of thousands of CPU cores. Similarly (Simonyan & Zisserman, 2014) has employed 11-19 layers and achieved excellent classification results on the ImageNet dataset. However, the increasing depth and width demands higher computational power. This high computational complexity is a major obstacle in porting the benefits of deep learning to resource limited devices. Further, the hotspot for optimization are the convolution layers, as most of the computations are conducted there. Therefore, many researchers have proposed ideas to accelerate deep networks for real-time inference Yu et al. (2012); Han et al. (2015b;a); Mathieu et al. (2013); Anwar et al. (2015b).

Network pruning is one promising technique that first learns a function with a sufficiently large sized network followed by removing less important connections Yu et al. (2012); Han et al. (2015b); Anwar et al. (2015b). This enables smaller networks to inherit knowledge from the large sized predecessor networks and exhibit a comparable level of performance. The works of Han et al. (2015b;a) introduce fine grained sparsity in a network by pruning scalar weights. Due to unstructured sparsity, the authors employ compressed sparse row/column (CSR/CSC) for sparse representation. Thus the fine grained irregular sparsity cannot be easily translated into computational speedups.

Sparsity in a deep convolutional neural network (CNN) can be induced at various levels. Figure 1 shows four pruning granularities. At the coarsest level, a full hidden layer can be pruned. This is shown with a red colored rectangle in Fig. 1(a). Layer wise pruning affects the depth of the network and a deep network can be converted into a shallow network. Increasing the depth improves the network performance and layer-wise pruning therefore demand intelligent techniques to mitigate the

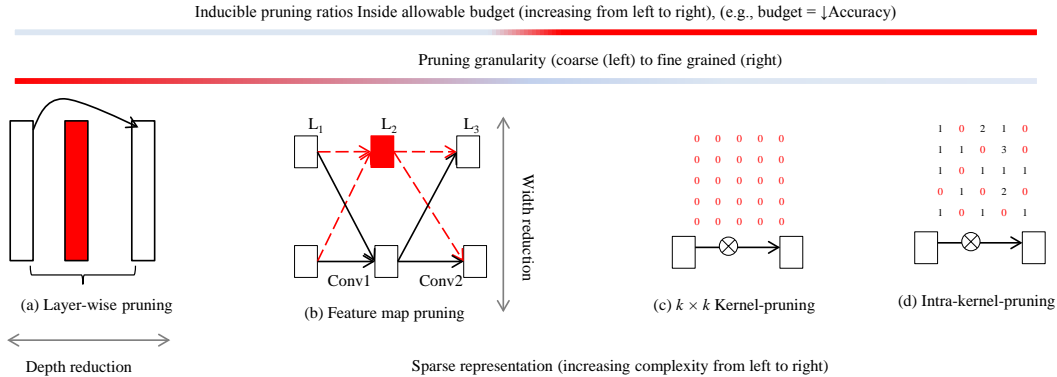

Figure 1: (a-d) shows four possible pruning granularities. The proposed work is focussed on the (b) feature map and (c) kernel pruning for simple sparse represenation. It can be observed that for the depicted architecture in Fig. (b), four convolution kernels are pruned.

performance degradation. The next pruning granularity is removing feature maps Polyak & Wolf (2015); Anwar et al. (2015b). Feature map pruning removes a large number of kernels and may degrade the network performance much. We therefore may not achieve higher pruning ratios with this granularity. For the depicted architecture in Fig. 1 (b)., pruning a single feature map, removes four kernels. Feature map pruning affects the layer width and we directly obtain a thinner network and no sparse representation is needed. Kernel pruning is the next pruning granularity and it prunes $k \times k$ kernels. It is neither too fine nor too coarse and is shown in Fig. 1(c). Kernel pruning is therefore a balanced choice and it can change the dense kernel connectivity pattern to a sparse one. Each convolution connection involves $W \times H \times k \times k$ multiply and accumulate (MAC) operations where *W, H* and *k* represents the feature map width, height and the kernel size, respectively.

Further the sparse representation for kernel pruning is also very simple. A single flag is enough to represent one convolution connection. Generally, the pruning techniques induce sparsity at the finest granularity by removing scalar weights. This sparsity can be induced in much higher rates but high pruning ratios do not directly translate into computational speedups in VLSI or parallel computer based implementations Han et al. (2015b). Figure 1(d) shows this with red colored zeroes in the kernel. Further Fig. 1 summarizes the relationship between three related factors: the pruning granularities, the pruning ratios and the sparse representations. Coarse pruning granularities demand very simple sparse representation but higher pruning ratios are comparatively difficult to achieve. Similarly fine grained pruning granularities can achieve higher pruning ratios but the sparse representation is more complicated. The proposed work therefore prunes feature maps and kernels in a network. Experimental evaluations show that better pruning results can be achieved when a network is pruned with both granularities successively.

An important contribution of this work is proposing a simple and generic strategy for the selection of pruning masks. Finding pruning candidates is an important and difficult prob-

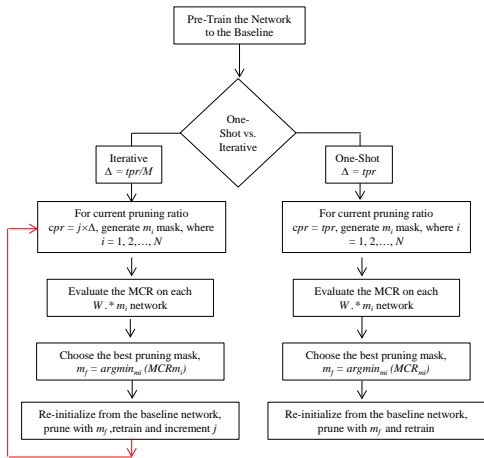

Figure 2: This figure compares the iterative and one-shot pruning. $tpr$ and $cpr$ represents the target and current pruning ratio respectively. The iterative pruning Han et al. (2015b) gradually achieves the target pruning ratio in $M$ steps of $\Delta$ size each, while the $\Delta = tpr$ for one-shot pruning. This work adopts the one-shot pruning approach.

lem. Generally, in the literature granularity specific pruning strategies are reported Han et al. (2015b); Li et al. (2016). (Anwar et al., 2015b) have developed a particle filtering approach, where the sequential importance resampling is employed. The proposed strategy randomly generates $N$ pruning masks, evaluates the importance of each mask with the validation set, selects the best mask having the $argmin_{m_i}(MCR_{m_i})$, prunes and retrains the network Yu et al. (2012). It is important to mention here that the pruning can be conducted in one-shot or iteratively. This difference is shown in Fig. 2. For a target pruning ratio ($tpr$), the iterative process gradually increases sparsity and repeats the process $M$ times. On the other hand, the one-shot pruning induces the target pruning ratio in one step. We employ one-shot pruning as the retraining after pruning consumes much time. Thus, the one shot pruning is much more efficient in terms of the optimization time. We show experimentally that the proposed algorithm can select better pruning candidates compared to other methods. Further, our approach is not computationally expensive as it involves $N$ random evaluations on the small sized validation set.

The rest of the paper is organized as follows. Section 2 provides detailed explanations on the pruning candidate selection. Section 3 discusses the two pruning granularities while Section 4 presents the experimental results. In Section 5, recent related works are revisited. We finally conclude the discussion in Section 6 and add the future research dimensions for this work.

## 2 PRUNING CANDIDATE SELECTION

Pruning reduces the number of network parameters and inevitably degrades the classification performance. The pruning candidate selection is therefore of prime importance. For a specific pruning ratio, we search for the best pruning masks which afflicts the least adversary on the pruned network. Indeed retraining can partially or fully recover the pruning losses, but the lesser the losses, the more plausible is the recovery Mishkin & Matas (2015). Further small performance degradation also means that the successor network has lost little or no knowledge of the predecessor network. If there are $M$ potential pruning candidates, the total number of pruning masks is $(2^M)$ and an exhaustive search is therefore infeasible even for a small sized network. We therefore propose a simple and greedy strategy for selecting pruning candidates.

We initialize a network with pre-trained parameters. These parameters may be learnt on the same or related problem. We randomly generate $N$ pruning masks and compute the misclassification rate (MCR) for each one. We then choose the best pruning mask with maximum accuracy on the validation set. Referring to the depicted architecture in Fig.4a, suppose we need to select feature map pruning candidates in layer $L_2$ and $L_3$ with $1/3$ pruning ratio. If $N = 4$, the following $N$ ordered pairs of feature maps may be randomly selected for $(L_2, L_3)$ : (1, 2), (2, 3), (3, 1), (1, 1). These combinations generate random paths in the network and we evaluate the validation set MCR through these routes in the network.

However, this further raises the question of how to approximate $N$. We analyze the relationship between pruning ratio and $N$ on three datasets and the results are reported in Fig. 3. This analysis is conducted for feature map pruning but is also applicable to other pruning granularities. It can be observed from Fig. 3a and 3c, that for higher pruning ratios, bigger value of $N$ is beneficial as it results in better pruning candidate selection. Moreover, for the pruning ratio of no more than 40%, $N = 50$ random evaluations generate good selections. For lower pruning ratios, retraining is also more likely to compensate the losses as the non-pruned parameters may still be in good numbers. The computational cost of this technique is not much as the evaluation is conducted on the small sized validation set. By observing Fig. 3a and 3c., we propose that the value of $N$ can be estimated initially and later used in several pruning passes. The plots in Fig. 3b and 3d show the pre-retraining distribution of $N$ random masks. Further, the plots in Fig. 3b and 3d, shows that the distributions are narrow for small pruning ratios.

We further analyze the effect of retraining on the pruning mask selection. We prune a network with several masks and retrain each pruned network. As several networks needs to be pruned and retrained many times, we experiment with a small network where the architecture is reported like this: $32(C5) - MP2 - 64(C5) - MP2 - 64(C5) - 64FC - 10Softmax$. The network is trained with the CIFAR-10 dataset (40,000 training samples) without any data augmentation and batch normalization. The network achieves the baseline performance of 26.7% on the test set. The results are reported in Fig. 4d, where the pre and post-retraining network performance is shown on

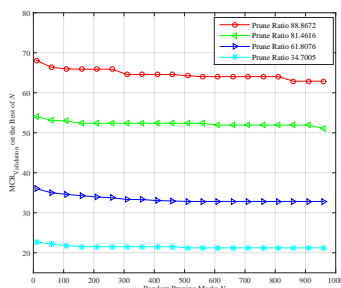

(a) Best of $N$ masks for CIFAR10 $CNN_{small}$

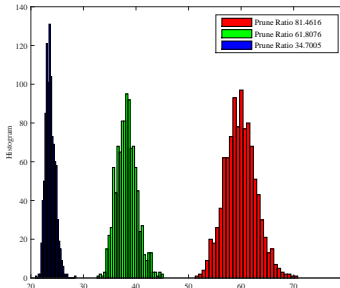

(b) Distribution of $N$ masks for CIFAR10 $CNN_{small}$

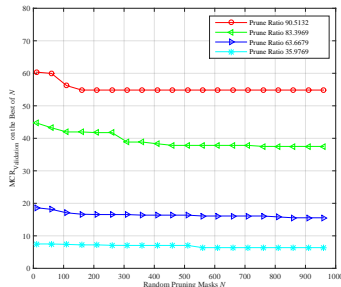

(c) Best of $N$ masks for $CNN_{SVHN}$

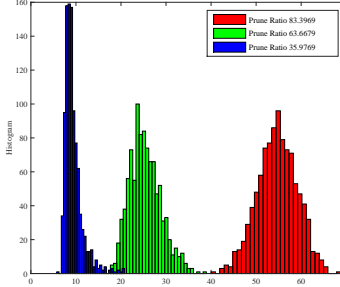

(d) Distribution of $N$ masks for $CNN_{SVHN}$

Figure 3: The network architectures are reported in Table 1. The networks are feature map pruned to generate the pre-retraining plots. Figure (a, c) compares the best candidate selected out of $N$ random combinations for various pruning ratios. The distribution of $N$ random evaluations is shown in Fig. (b, d). We can observe that it resembles a Gamma distribution. Further, for higher pruning ratios, the distribution resembles a bell-shaped curve. Analyzing Fig. (a,c) with Fig. (b,d), we infer that bigger $N$ may be beneficial for higher pruning ratios.

the $x$ and $y$ axis, respectively. Further, we superimpose a least-squares (LS) line fit to each of the scatter plot. It can be observed that the slope of the LS line decreases for higher pruning ratios. We infer that for high pruning ratios, the final network performance is dictated by the surviving number of effective parameters. It can be observed that the overall distribution is noisy. However, in general, the pre-retraining least adversarial pruning masks perform better after retraining. In the rest of this work, we therefore use the pre-retraining best mask for pruning the network.

We further compare this method with the weight sum criterion proposed in Li et al. (2016) and shown in Fig. 4a. The set of filters or kernels from the previous layer constitute a group. This is shown with the similar color in Fig. 4a. According to Li et al. (2016), the absolute sum of weights determine the importance of a feature map. Suppose that in Fig.4a, the Layer $L_2$ undergoes feature map pruning. The weight sum criterion computes the absolute weight sum at $S_1$, $S_2$ and $S_3$. If we further suppose that the pruning ratio is $1/3$, then the $min(S_1, S_2, S_3)$ is pruned. All the incoming and outgoing kernels from the pruned feature map are also removed. We argue that the sign of a weight in kernel plays important role in well-known feature extractors and therefore this is not a good criterion.

We compare the performance of the two algorithms and Fig. 4b and 4c shows the experimental results. These results present the network status before any retraining is conducted. We report the performance degradation in the network classification against the pruning ratio. From Fig. 4b and 4c, we can observe that our proposed method outperforms the weight sum method particularly for higher pruning ratios. The *best of N Pruning masks* strategy evaluates pruning candidates in combinations and provides a holistic view. The criterion in Li et al. (2016) evaluates the importance of a pruning unit in the context of a single layer while our proposed approach evaluates several paths through the network and selects the best one. The combinations work together and matter more instead of individual units. Further, our proposed technique is generic and can be used for any pruning granularity: feature map, kernel and intra-kernel pruning.

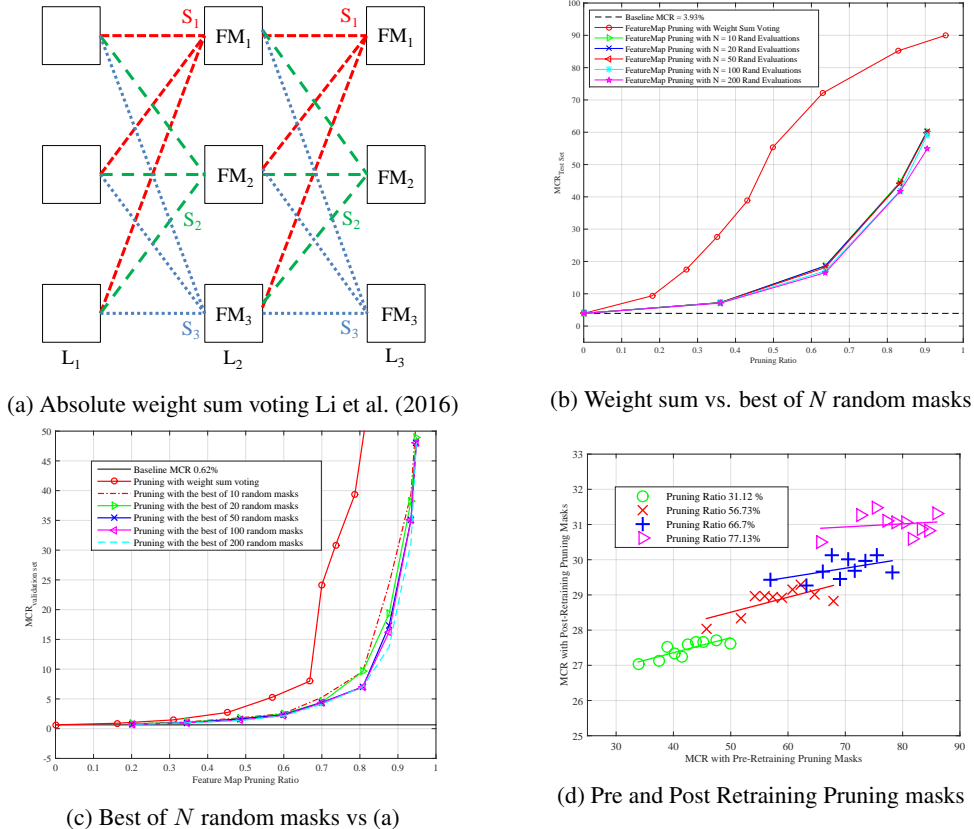

(a) Absolute weight sum voting Li et al. (2016)

(b) Weight sum vs. best of $N$ random masks

(c) Best of $N$ random masks vs (a)

(d) Pre and Post Retraining Pruning masks

Figure 4: (a) This figure explains the idea presented in Li et al. (2016) and shows three layers, $L1$, $L2$ and $L3$. All the filters/kernels from previous layer to a feature map constitute one group which is shown with similar color. The $S1$,$S2$ and $S3$ is computed by summing the absolute value of all the weights in this group. (b) The comparison of the proposed method with the absolute weight sum method is shown here for the $CNN_{SVHN}$. It can be observed that our proposed method inflicts lesser adversary on the network for different pruning ratios. (d) In this plot, we prune a CNN network with various masks and compare their pre and post retraining performance. It can be observed that on the average, pre-retraining masks perform better after retraining.

## 3 FEATURE MAP AND KERNEL PRUNING

In this section we discuss feature map and kernel pruning granularities. For a similar sized network, we analyze the achievable pruning ratios with feature map and kernel pruning. In terms of granularity, feature map pruning is coarser than kernel pruning. Feature map pruning does not need any sparse representation and the pruned network can be implemented in a conventional way, convolution lowering Chellapilla et al. (2006) or convolution with FFTs Mathieu et al. (2013). The proposed work analyzes the unconstrained kernel and feature map pruning. Pruning a feature map eliminates all the incoming and outgoing kernels because the outgoing kernels are no more meaningful.

Kernel pruning is comparatively finer. The dimension and connectivity pattern of 2D kernels determine the computing cost of a convolutional layer. The meshed fully connected convolution layers increases this cost and can hinder the real-time inference. In LeNet LeCun et al. (1998), the second convolution layer has $6 \times 16$ feature maps and the kernel connectivity has a fixed sparse pattern. With kernel pruning, we learn this pattern and convert the dense connectivity to sparse one. Kernel pruning zeroes $k \times k$ kernels and is neither too fine nor too coarse. Kernel level pruning provides a balance between fine-grained and coarse-grained pruning. It is coarser than the intra-kernel sparsity and finer than the feature map pruning. Depending on the network architecture, kernel pruning may achieve good pruning ratios at very small sparse representation and computational cost. Each convolution connection represents one convolution operation which involves $width \times height \times k \times k$

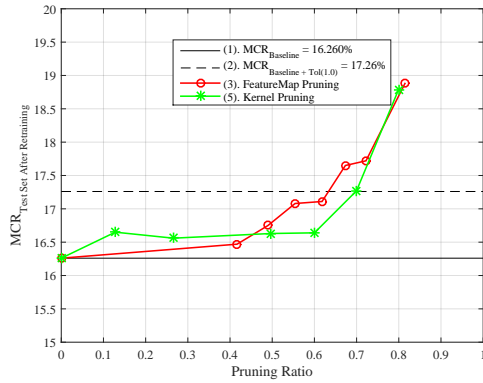

(a) Feature map and kernel pruning of CIFAR-10 $CNN_{small}$

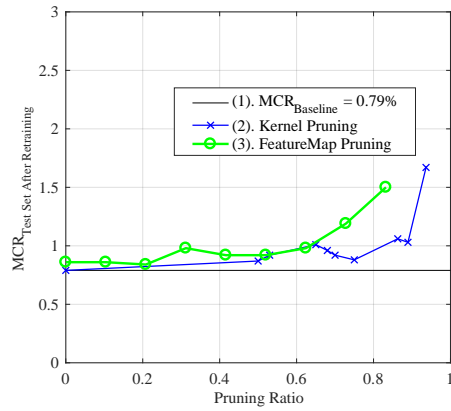

(b) MNIST feature map and kernel pruning

Figure 5: Figure (a) and (b) shows feature map and kernel pruning of two networks: $CNN_{CIFAR-10.small}$ and $CNN_{MNIST2}$. The corresponding network architectures are reported in Table 1. The network can be pruned by more than 50% with very small degradation in performance. Further, due to finer nature, the kernel pruning may inflict lesser adversary on the network performance.

MAC operations. We first select pruning candidates with the criterion outlined in Section 2. The pruned network is then retrained to compensate for the losses incurred due to pruning. Figure 5a and 5b show that depending on the network architecture, kernel pruning may achieve higher pruning ratio than feature map pruning due to finer granularity. As the sparse granularities are coarse, a generic set of computing platform can benefit from it. One disadvantage of the unconstrained kernel pruning is that the convolution unrolling technique cannot benefit from it Chellapilla et al. (2006). However, customized VLSI implementations and FFT based convolutions do not employ convolution unrolling. Mathieu et al. (2013), have proposed FFT based convolutions for faster CNN training and evaluation and the GPU based parallel implementation showed very good speedups. As commonly known that the $IFFT(FFT(kernel) \times FFT(featuremap)) = kernel * featuremap$, the kernel level pruning can relieve this task. Although the kernel size is small, massive reusability of the kernels across the mini-batch enables the use of FFT. The FFT of each kernel is computed only once and reused for multiple input vectors in a mini-batch. In a feed-forward and backward path, the summations can be carried in the FFT domain and once the sum is available, the IFFT can

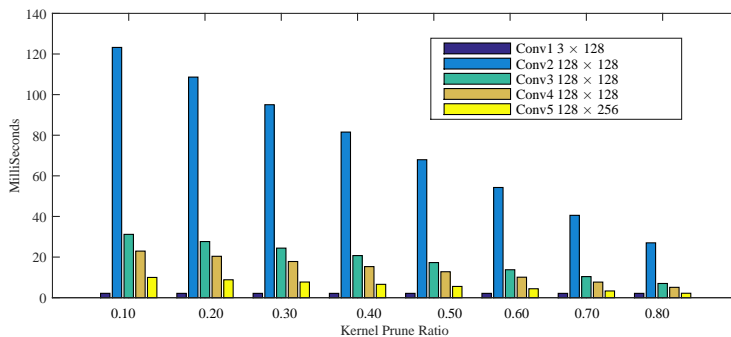

(a) Profiling kernel pruning

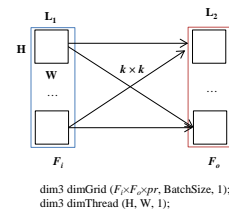

(b) Custom GPU kernel for convolutions

Figure 6: (a) This figure shows the profiling results for kernel pruning with a customized GPU implementation. It can be observed that the kernel pruning reduces the execution time. The experiment is conducted with the CIFAR-10 CNN. In (b), $F_i$ and $F_o$ shows the input and output feature maps, while $pr$ represents the pruning ratio. The GPU function scheduler shows that the call is only for non-masked kernels.

Table 1: Specifications of the three CIFAR-10 networks

| Network | Architecture | Baseline MCR(%) | Data Augmentation |
|---|---|---|---|
| $CNN_{MNIST1}$ | $16(C5) - 32(C5) - 64(C5) - 120 - 10$ | 0.62 | NO |
| $CNN_{MNIST2}$ | $6(C5) - 16(C5) - 120(C5) - 84 - 10$ | 0.79 | NO |
| $CNN_{CIFAR10.small}$ | $2 \times 128C3 - MP2 - 2 \times 128C3 - MP2 - 2 \times 256C3 - 256FC - 10Softmax$ | 16.6 | NO |
| $CNN_{CIFAR10.large}$ | $2 \times 128C3 - MP2 - 2 \times 256C3 - MP2 - 2 \times 256C3 - 1 \times 512C3 - 1024FC - 1024FC - 10Softmax$ | 9.41 | YES |
| $CNN_{SVHN}$ | $(2 \times 64C3) - MP2 - (2 \times 128C3) - MP2 - (2 \times 128C3) - 512FC - 512FC - 10Softmax$ | 3.5 | NO |
| $CNN_{CIFAR100}$ | $(2 \times 128C3) - MP2 - (2 \times 128C3) - MP2 - (2 \times 256C3) - 256C3 - 512FC - 10Softmax$ | 33.65 | YES |

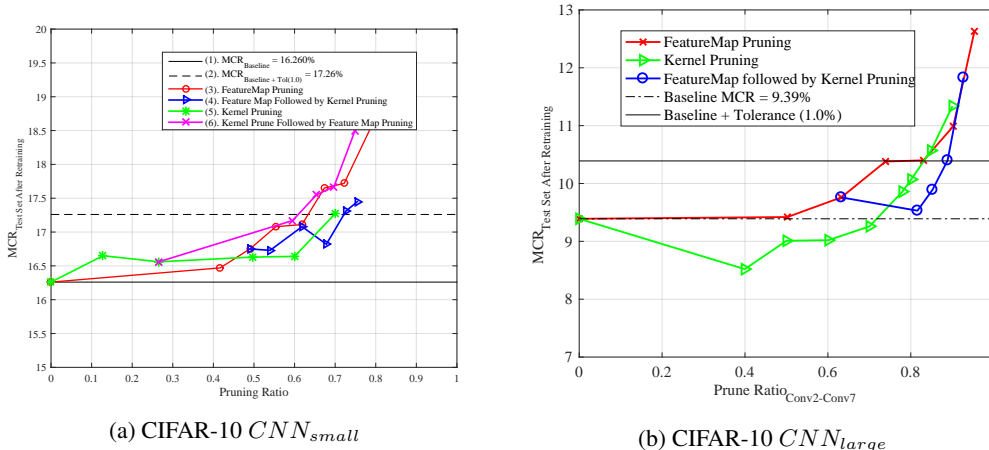

(a) CIFAR-10 $CNN_{small}$ (b) CIFAR-10 $CNN_{large}$

Figure 7: The combinations of feature map and kernel pruning is reported here. Figure (a) and (b) provides pruning results for the $CNN_{CIFAR10.small}$ and $CNN_{CIFAR10.large}$ networks. It can be observed from both figure, that more sparsity can be induced in the network by indcuing sparsity with two granularities.

be performed Mathieu et al. (2013). Similarly, a customized VLSI based implementation can also benefit from the kernel level pruning. If the VLSI implementation imposes a constraint on the pruning criterion, such as the fixed number of convolution kernels from the previous to the next layer, the pruning criterion can be adapted accordingly. In the next Section, we report and discuss the experimental results in detail. As the commonly available libraries do not support masked convolutions, we therefore profile kernel pruning with customized GPU functions. We call the GPU function only for the non-pruned convolution kernels and pass the appropriate indices. It can be observed that fewer number of convolutions will reduce the required number of GFLOPs. Howevr, we conjecture that the true benefit of kernel pruning can be obtained with FFT based masked convolution.

## 4 EXPERIMENTAL RESULTS

In this section, we present detailed experimental results with the CIFAR-10 and SVHN datasets Krizhevsky & Hinton (2009). We experiment on three image classification problems and induce sparsity feature map and kernel wise. We also prune one network with more than one pruning granularity in combinations. During training and pruning, we use the stochastic gradient descent (SGD) and batch normalization Ioffe & Szegedy (2015). As elaborated in Section 1, we do not prune the network in small steps, and instead one-shot prune the network for a given pruning ratio followed by retraining. The experimental results are reported in the corresponding subsections.

### 4.1 CIFAR-10

The CIFAR-10 dataset includes samples from ten classes: airplane, automobile, bird, cat, deer, dog, frog, horse, ship and truck. The training set consists of 50,000 RGB samples and we allocate 20% of these samples as validation set. Test set contains 10,000 samples and each sample has $32 \times 32 \times RGB$ resolution. We evaluate the proposed pruning granularities with two networks. $CNN_{CIFAR10.small}$ and $CNN_{CIFAR10.large}$. $CNN_{CIFAR10.small}$ has six convolution and two overlapped max pooling layers. We report the network architecture with an alphanumeric string as

Table 2: Feature map and kernel level pruning (75%) in $CNN_{CIFAR10.small}$

| Feature Maps | Pruned Feature Maps | Feature Maps Prune Ratio | Pruned Kernels (%) | Conv Connections | Kernel Prune Ratio (%) |
|---|---|---|---|---|---|
| $Conv2(128 \times 128)$ | $128 \times 89$ | 30.5 | 27306/9 = 3034 | 11392 | 3034/11392 = 26.6 |
| $Conv3(128 \times 128)$ | $89 \times 89$ | 51.5 | 18702/9 = 2078 | 7921 | 2078/7921 = 26.2 |
| $Conv4(128 \times 128)$ | $89 \times 89$ | 51.5 | 18702/9 = 2078 | 7921 | 2078/7921 = 26.2 |
| $Conv5(128 \times 256)$ | $89 \times 179$ | 51.4 | 37881/9 = 4209 | 15931 | 4209/15931 = 26.4 |
| $Conv6(256 \times 256)$ | $179 \times 179$ | 51.1 | 76851/9 = 8539 | 32041 | 8539/32041 = 26.6 |

reported in Courbariaux et al. (2015) and outlined in Table 1. The $(2 \times 128C3)$ represents two convolution layers with each having 128 feature maps and $3 \times 3$ convolution kernels. $MP2$ represents $3 \times 3$ overlapped max-pooling layer with a stride size of 2. We pre-process the original CIFAR-10 dataset with global contrast normalization followed by zero component analysis (ZCA) whitening.

The $CNN_{CIFAR10.large}$ has seven convolution and two max-pooling layers. Further, online data augmentations are employed to improve the classification accuracy. We randomly crop $28 \times 28 \times 3$ patches from the $32 \times 32 \times 3$ input vectors. These cropped vectors are then geometrically transformed randomly. A vector may be flipped horizontally or vertically, rotated, translated and scaled. At evaluation time, we crop patches from the four corners and the center of a $32 \times 32 \times 3$ patch and flip it horizontally. We average the evaluation on these ten $28 \times 28 \times 3$ patches to decide the final label. Due to larger width and depth, the $CNN_{CIFAR10.large}$ achieves more than 90% accuracy on the CIFAR-10 dataset. The $CNN_{CIFAR10.small}$ is smaller than $CNN_{CIFAR10.large}$ and trained without any data augmentation. The $CNN_{CIFAR10.small}$ therefore achieves 84% accuracy.

### 4.1.1 FEATURE MAP AND KERNEL LEVEL PRUNING

After layer pruning, feature map pruning is the 2nd coarsest pruning granularity. Feature map pruning reduces the width of a convolutional layer and generates a thinner network. Pruning a single feature map, zeroes all the incoming and outgoing weights and therefore, higher pruning ratios degrade the network classification performance significantly. Feature map pruning for the $CNN_{CIFAR10.small}$ is shown in Fig. 5a with a circle marked red colored line. The sparsity reported here is for Conv2 to Conv6. We do not pruned the first convolution layer as it has only $3 \times 128 \times (3 \times 3) = 3456$ weights. The horizontal solid line shows the baseline MCR of 16.26% whereas the dashed line shows the 1% tolerance bound. Training the network with batch normalization Ioffe & Szegedy (2015) enables us to directly prune a network for a target ratio, instead of taking small sized steps. With a baseline performance of 16.26%, the network performance is very bad at 80% feature map pruning. We can observe that 62% pruning ratio is possible with less than 1% increase in MCR. The $CNN_{CIFAR10.small}$ is reduced to $(128C3 - 83C3)$-MP3-$(83C3 - 83C3)$-MP3-$(166C3 - 166C3)$-256$FC$-10$Softmax$. As pruning is only applied in Conv2 to Conv6, therefore the Figure 5a pruning ratios are computed only for these layers.

For the same network, we can see that kernel level pruning performs better. We can achieve 70% sparsity with kernel level pruning. This is attributed to the fact that kernel pruning is finer and hence it achieves higher ratios. Further kernel pruning may ultimately prune a feature map if all the incoming kernels are pruned. However at inference time, we need to define the kernel connectivity pattern which can simply be done with a binary flag. So although the sparse representation is needed, it is quite simple and straightforward. Experimental results confirm that fine grained sparsity can be induced in higher rates. We achieved 70% kernel wise sparsity for Conv2 - Conv6 and the network is compressed with very simple sparse representation.

### 4.1.2 COMBINATIONS OF KERNEL AND FEATURE MAP PRUNING

In this section we discuss the various pruning granularities applied in different combinations. We first apply the feature map and kernel pruning to the $CNN_{CIFAR10.small}$ network in different orders. With feature map pruning, we can achieve 60% sparsity under the budget of 1% increase in MCR. But at this pruning stage, the network learning capability is affected much. So we take a 50% feature map pruned network, where the $CNN_{CIFAR10.small}$ is reduced to $(128C3 - 89C3)$-MP3-$(89C3 - 89C3)$-MP3-$(179C3 - 179C3)$-256$FC$-10$Softmax$. As pruning is only applied to $Conv2 - Conv6$, therefore in Fig. 5a., pruning ratios are computed only for these layers. This network then undergoes kernel level pruning. The blue rectangle line in Figure 7a shows the pruning

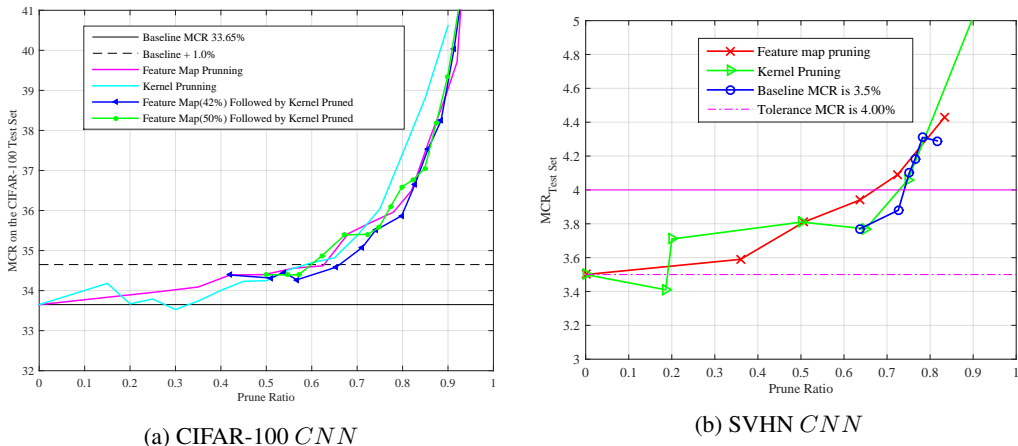

(a) CIFAR-100 $CNN$ (b) SVHN $CNN$

Figure 8: The pruning plots for 100 class classification problem is reported in (a). It can be observed that this network can be pruned by more than 60% with very small degradation in the network performance. Figure (b) shows the pruning results for the $CNN_{SVHN}$. It can be observed that more than 70% sparsity can be induced in the network while the network accuracy still remains above 96%.

results. We achieve the best pruning results in this case and the final pruned network is reported in detail in Table 2. Overall we achieve more than 75% pruning ratio in the final pruned network.

We further conducted experiments on the $CNN_{CIFAR10.large}$ and the corresponding plots are shown in Fig. 7b. The $CNN_{CIFAR10.large}$ is much wider and deeper than the CNNsmall as reported in Table 1. Therefore there are more chances of redundancy and hence more room for pruning. Further we observe similar trends as $CNN_{CIFAR10.small}$ where the kernel pruning can be induced in higher ratios compared to the feature map pruning. When the kernel pruning is applied to the feature map pruned network, we can achieve more than 88% sparsity in the $Conv2 - Conv7$ of the $CNN_{CIFAR10.large}$ network. This way we show that our proposed technique has good scalability. These results are in conformity to the resiliency analysis of fixed point deep neural networks Sung et al..

## 4.2 CIFAR-100

The CIFAR-100 dataset has 50,000 images classified into 100 fine and 20 coarse labels. The dataset has 50,000 training and 10,000 test set images. The hundred class classification problem of CIFAR-100 has 500 images for each class. We construct a validation set for learning rate scheduling during training. The validation set is constructed with 100 samples for each class from the training set. This way we are left with 400 samples per class for training. We train the network with 40,000 images with data augmentation and batch normalization Ioffe & Szegedy (2015). We obtain a baseline accuracy of 33.65% on the CIFAR-100 test set with a VGG styled network. The network architecture is reported in Table 1 as $CNN_{CIFAR100}$.

The pruning plots for this dataset are provided in Fig. 8a. It can be observed that around 60% of the network parameters can be pruned with less than 1% (absolute) increase in the network performance. Moreover, pruning in combinations further improve the pruning ratios. Thus the lessons learnt generalize well to other datasets.

## 4.3 SVHN

The SVHN dataset consists of $32 \times 32 \times 3$ cropped images of house numbers [Netzer et al. 2011] and bears similarity with the MNIST handwritten digit recognition dataset [LeCun et al. 1998]. The classification is challenging as more than one digit may appear in sample and the goal is to identify a digit in the center of a patch. The dataset consists of 73,257 digits for training, 26,032 for testing and 53,1131 extra for training. The extra set consists of easy samples and may augment the

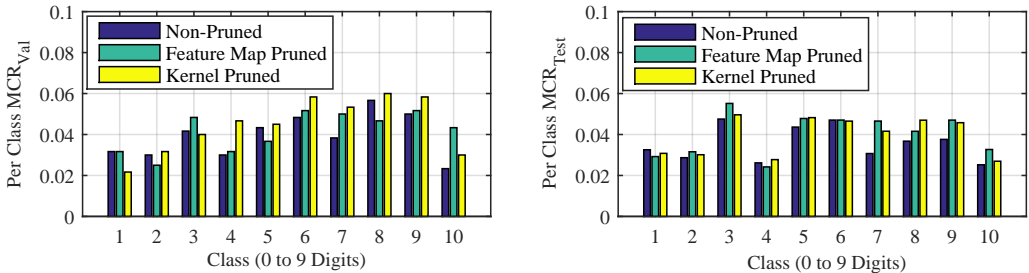

Figure 9: This figure shows the per class MCR for the original, feature map, and kernel pruned networks. It can be observed that the per class error does not vary much in the pruned networks. This shows that the pruning method is not biased towards a specific class. The feature map pruned network has 63.67% sparsity with $MCR_{Test} = 3.84\%$, $MCR_{Val} = 4.16\%$. The kernel pruned network has 65.01% sparsity with $MCR_{Test} = 3.77\%$, $MCR_{Val} = 4.45\%$. The sparsity are computed for Conv2-Conv6.

training set. We generate a validation set of 6000 samples which consists of 4000 samples from the training set and 2000 samples from the extra [Sermanet et al. 2012]. The network architecture is reported like this: $(2 \times 64C3)$-MP2- $(2 \times 128C3)$-MP2-$(2 \times 128C3)$-512FC-512FC-10Softmax. This network is trained with batch normalization and we achieve the baseline MCR of 3.5% on the test set. The corresponding pruning plots are reported in Fig. 8b. We can observe a similar trend where kernels can be pruned by a bigger ratio compared to feature maps. More than 70% pruning ratio can be implemented in the reported network. Thus we show that the lessons learnt generalize well on various datasets.

There can be a concern that pruning may decrease the accuracy of the original network when it is deployed in the field for run time classification. For a specific problem domain, the test set is used as a proxy for the future unseen data. We argue that to some extent, this question can be answered by comparing the per class error for the original and pruned networks. This way we can see whether the pruned network is biased towards a specific class. To analayze this, we computed the per class error with the $CNN_{SVHN}$ network as reported in Table 1. The results are reported in Fig. 9. It can be observed that the per class error for both validation and test set do not vary significantly. We therefore infer that the pruning and retraining process is a promising technique for complexity reduction.

## 5 RELATED WORK

In the literature, network pruning has been studied by several researches Han et al. (2015b;a); Yu et al. (2012); Castellano et al. (1997); Collins & Kohli (2014); Stepniewski & Keane (1997); Reed (1993). Collins & Kohli (2014) have proposed a technique where irregular sparsity is used to reduce the computational complexity in convolutional and fully connected layers. However they have not discussed how the sparse representation will affect the computational benefits. The works of Han et al. (2015b;a) introduce fine-grained sparsity in a network by pruning scalar weights. If the absolute magnitude of any weight is less than a scalar threshold, the weight is pruned. This work therefore favors learning with small valued weights and train the network with the L1/L2 norm augmented loss function. Due to pruning at very fine scales, they achieve excellent pruning ratios. However this kind of pruning results in irregular connectivity patterns and demand complex sparse representation for computational benefits. Convolutions are unrolled to matrix-matrix multiplication in Chellapilla et al. (2006) for efficient implementation. The work of Lebedev & Lempitsky (2015) also induce intra-kernel sparsity in a convolutional layer. Their target is efficient computation by unrolling convolutions as matrix-matrix multiplication. Their sparse representation is not also simple because each kernel has an equally sized pruning mask. A recently published work propose sparsity at a higher granularity and induce channel level sparsity in a CNN network for deep face application Polyak & Wolf (2015). The work of Castellano et al. (1997); Collins & Kohli (2014); Stepniewski & Keane (1997); Reed (1993) utilize unstructured fine grained sparsity in a neural network. Fixed

point optimization for deep neural networks is employed by Anwar et al. (2015a); Hwang & Sung (2014); Sung et al. for VLSI based implementations. The reference work of Anwar et al. (2015b) analyzed feature map pruning with intra-kernel strided sparsity. To reduce the size of feature map and kernel matrices, they further imposed a constraint that all the outgoing kernels from a feature map must have the same pruning mask. In this work, we do not impose any such constraint and the pruning granularities are coarser. We argue that this kind of sparsity is useful for VLSI and FFT based implementations. Moreover we show that the best pruning results are obtained when we combine feature map and kernel level pruning.

## 6 Concluding Remarks

In this work, we proposed feature map and kernel pruning for reducing the computational complexity of deep CNN. We have discussed that the cost of sparse representation can be avoided with coarse pruning granularities. We demonstrated a simple and generic algorithm for selecting the best pruning mask from a random pool. We showed that the proposed approach adopts a holistic approach and performs better than the other methods. Further, we adopted the efficient one-shot pruing approach as the iterative retraining consumes much time. We conducted experiments with several benchmarks and networks and showed that the proposed technique has good scalability.

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
