# Peer review of "Coarse Pruning of Convolutional Neural Networks with Random Masks"

_ICLR 2017 — rejected_

[Official Review · AnonReviewer3 · rating 4 · confidence 4 · 15 Dec 2016 (modified: 21 Jan 2017)]
**Useful topic, but not very solid techinique**

This paper proposes two pruning methods to reduce the computation of deep neural network. In particular, whole feature maps and the kernel connections can be removed with not much decrease of classification accuracy. 
However, this paper also has the following problems. 
1)	The method is somehow trivial, since the pruning masks are mainly chosen by simple random sampling. The novelty and scalability are both limited. 
2)	Experiment results are mainly focused on the classification rate and the ideal complexity. As a paper on improving computation efficiency, it should include results on practical time consumption. It is very common that reducing numbers of operations may not lead to reduced computational time on a highly parallel platform (e.g., GPU). 
3)	It is more important to improve the computational efficiency on large-scale models (e.g., ImageNet classification network) than on small models (e.g., MNIST, CIFAR network). However, results on large-scale network is missing.
4)	(*Logical validity of the proposed method*) For feature map pruning, what if just to train reduced-size network is trained from scratch without transfer any knowledge from the pretrained large network? Is it possible to get the same accuracy? If so, it will simply indicate the hyper-parameter is not optimal for the original network. Experimental results are necessary to clarify the necessity of feature map pruning. 
Note that I agree with that a smaller network may be more generalizable than a larger network. 

----------------------------------------------

Comments to the authors's response:

Thanks for replying to my comments. 

1) I still believe that the proposed methods are trivial.
2) It is nice to show GPU implementation. Compared to existing toolboxes (e.g., Torch, Caffe, TensorFlow), is the implementation of convolution efficient enough?
3) Experiments on Cifar-100 are helpful (better than cifar-10), but it is not really large-scale, where speed-up is not so critical. ImageNet and Places datasets are examples of large-scale datasets.
4) The author did not reply to the question wrt the validity of the proposed methods. This question is critical.

[Official Review · AnonReviewer4 · rating 6 · confidence 3 · 15 Dec 2016 (modified: 23 Jan 2017)]
**N random trails to get best pruning?**

Summary: There are many different pruning techniques to reduce memory footprint of CNN models, and those techniques have different granularities (layer, maps, kernel or intra kernel), pruning ratio and sparsity of representation. The work proposes a method to choose the best pruning masks out to many trials. Tested on CIFAR-10, SVHN and MNIST.

Pros:
Proposes a method to choose pruning mask out of N trials. 
Analysis on different pruning methods.

Cons & Questions:
“The proposed strategy selects the best pruned network through N random pruning trials. This approach enables one to select pruning mask in one shot and is simpler than the multi-step technique.” How can one get the best pruning mask in one shot if you ran N random pruning trials? (answered)
Missing tests of the approach with bigger CNN: like AlexNet, VGG, GoogLeNet or ResNet. (extended to VGG ok)
Since reducing model size for embedded systems is the final goal, then showing how much memory space in MB is saved with the proposed technique compared with other approaches like Han et al. (2015) would be good.

Misc:
Typo in figure 6 a) caption: “Featuer” (corrected)

[Official Review · AnonReviewer1 · rating 5 · confidence 4 · 17 Dec 2016]
**Paper review**

This paper proposes a simple randomized algorithm for selecting which weights in a ConvNet to prune in order to reduce theoretical FLOPs when evaluating a deep neural network. The paper provides a nice taxonomy or pruning granularity from coarse (layer-wise) to fine (intra-kernel). The pruning strategy is empirically driven and uses a validation set to select the best model from N randomly pruned models. Makes claims in the intro about this being "one shot" and "near optimal" that cannot be supported: it is "N-shot" in the sense that N networks are generated and tested and there is no evidence or theory that the found solution is "near optimal."

Pros:
- Nice taxonomy of pruning levels
- Comparison to the recent weight-sum pruning method

Cons:
- Experimental evaluation does not touch upon recent models (ResNets) and large scale datasets (ImageNet)
- Paper is somewhat hard to follow
- Feature map pruning can obviously accelerate computation without specialized sparse implementations of convolution, but this is not the case for finer grained sparsity; since this paper considers fine-grained sparsity it should provide some evidence that introducing that sparsity can yield performance improvements

Another experimental downside is that the paper does not evaluate the impact of filter pruning on transfer learning. For example, there is not much direct interest in the tasks of MNIST, CIFAR10, or even ImageNet. Instead, a main interest in both academia and industry is the value of the learned representation for transferring to other tasks. One might expect pruning to harm transfer learning. It's possible that the while the main task has about the same performance, transfer learning is strongly hurt. This paper has missed an opportunity to explore that direction.

In summary, the proposed method is simple, which is good, but the experimental evaluation is somewhat incomplete and does not cover recent models and larger scale datasets.

[Public Comment · ICLR 2017 conference · 23 Jan 2017 (modified: 26 Jan 2017)]
**Reactions to author responses**

Dear reviewers,

can you please take a look at the responses by the authors and add a comment indicating that you have taken them into consideration?

Thanks!

[Public Comment · (anonymous) · 02 Feb 2017]
**Some related works**

1) Wen, Wei, et al. "Learning structured sparsity in deep neural networks." Advances in Neural Information Processing Systems. 2016.
2) Lebedev, Vadim, and Victor Lempitsky. "Fast convnets using group-wise brain damage." Proceedings of the IEEE Conference on Computer Vision and Pattern Recognition. 2016.
3) Alvarez, Jose M., and Mathieu Salzmann. "Learning the Number of Neurons in Deep Networks." Advances in Neural Information Processing Systems. 2016.

[Final Decision · Program Chairs · 06 Feb 2017]
**ICLR committee final decision**

Unfortunately this paper is not competitive enough for ICLR. The paper is focusing on efficiency where for the results to be credible it is of utmost importance to present experiments on large scale data and state-of-the-art models.
 I am afraid the reviewers were not able to take into account the latest drafts of the paper that were submitted in January, but those submissions came very late.